# Risks in Relation to Adopting Airbnb Accommodation: The Role of Fear of COVID-19

Mohamed Fathy Agina [1,*], Nadir Aliane [2], Osman El Sawy [3], Hazem Ahmed Khairy [4] and Sameh Fayyad [3]

1   Hotel Management Department, Higher Institute for Specific Studies, Heliopolis, Cairo 11771, Egypt
2   Management Department, College of Business Administration, King Faisal University,
    Al-Hassa 31982, Saudi Arabia
3   Hotel Studies Department, Faculty of Tourism and Hotels, Suez Canal University, Ismailia 41522, Egypt
4   Hotel Management Department, Faculty of Tourism and Hotels, University of Sadat City,
    Sadat City 32897, Egypt
*   Correspondence: dr.mohamed.fathy@fa-hiss.edu.eg

**Abstract:** This study examined the impacts of perceived Airbnb risks, i.e., performance, financial, physical, social, time, and psychological risks on tourists' intention to use Airbnb. The study also explored the moderating effect of the fear of COVID-19 on the relationship between perceived Airbnb risks and the tourist's intention to use Airbnb. The data was collected from 248 customers of Airbnb using a survey approach. The data were analyzed utilizing the Smart PLS V.4. The PLS-SEM results revealed that Airbnb (physical, psychological, time, performance, financial and social risks) had a significant negative effect on the intention to use Airbnb. On the other side, the fear of COVID-19 acted as a moderator between Airbnb's physical, psychological, and social risks and the intention to use Airbnb, indicating that customers tend to tolerate time and performance risks when having a high level of fear of COVID-19 and prefer to use Airbnb regardless of them. This shift in customer behavior towards customers' intention to use Airbnb in light of the fear of COVID-19 gives Airbnb an edge that should be exploited via remedying other risks. It was suggested that the Airbnb hosts' awareness of the importance of Airbnb in the context of tourism in Egypt should be enhanced. Additionally, a legislative framework should govern Airbnb transactions to secure tourists in dealing with Airbnb hosts. Professionalism in providing Airbnb services should also be adopted. Future studies in the context of the current subject could conduct multi-group analyses according to different types of Airbnb accommodation and use a larger sample size.

**Keywords:** COVID-19; perceived risks; Airbnb; accommodation; sharing economy; Egypt





## 1. Introduction

Airbnb provides a way to rent short-term lodgings ranging from individual rooms and complete homes to shared rooms [1]. Currently, Airbnb provides temporary accommodation facilities to travelers in more than 100,000 cities located in 220 countries [2]. Before the COVID-19 epidemic, Airbnb could host around 500 million visitors, with an average of 2 million visitors per night [3]. Although the great valuations and the quick expansion of the sharing economy, various Airbnb-related threats, and risks, such as some crimes and accidents including robbery, sexual assault, and homicide, persist [4]. Moreover, these risks have been exacerbated by the spread of the COVID-19 pandemic [5].

The COVID-19 pandemic affected tourism and hospitality negatively and, consequently, Airbnb [6]. However, it is challenging for the authorities to obtain baseline data from which to identify the negative consequences of COVID-19 on Airbnb stakeholders. Airbnb stakeholders are not immune to the COVID-19 pandemic. Due to pandemic-related travel restrictions throughout the world, Airbnb reservations have drastically declined [7]. Furthermore, due to their restricted entitlement to different forms of government support,

the effects of COVID-19 on Airbnb hosts were seriously staggering [5]. Despite the resumption of tourism flow, the combination of a continuously increasing number of infections, deaths, and the absence of adequate prevention and treatment has caused excessive levels of fear among consumers [8].

The Airbnb protocols necessitate trust between guests and hosts beyond the transaction stage [9]. Nonetheless, COVID-19 has sowed mistrust among Airbnb stakeholders and introduced new problems in terms of customer service and technology [7]. Also, by nature, Airbnb is considered risky, as customers pay to stay with generally unlicensed strangers instead of relying on well-established lodging with a well-known global brand [10]. Conversely, considering the fear of the COVID-19 pandemic, Airbnb may be viewed as a more attractive option for customers because it reduces the level of interaction among the customers. Consequently, it seems the safer option when considering the spread of the virus [11]. In addition, its low price makes it less risky. Consumer shows resistance toward expensive purchasing during uncertain situations [12]. Yet, little is known about customers' perceptions of risks during the pandemic. Here, scholars argue that if customer Airbnb risk perception is high because of the fear of the COVID-19 pandemic, this directly negatively affects the intention to use Airbnb facilities and services [13].

Various scholars focused on studying the decision-making process of Airbnb's customers on some factors such as sustainability and authentic experience from the local community. However, the role of the perceived risks during the process still needs to be fully considered [14,15], especially during the COVID-19 pandemic and in developing countries. Therefore, our study seeks to bridge this gap by depending on Jacoby and Kaplan's six constructs to examine the relationship between Airbnb risks (i.e., performance, financial, physical, psychological, social, and time loss risks) and tourist's intention to use Airbnb and the moderating effect of the fear of COVID-19 to gain insights from consumers' behavior towards Airbnb use in the wake of COVID-19 to administer them well.

## 2. Literature Review

### 2.1. Airbnb Theoretical Background

In the sharing economy accommodation, people mostly share services and products with each other without cost or with a fee [16]. Many different organizations are involved in the sharing economy, including numerous in the travel sector, such as Outdoorsy [17], which enables people to reserve recreational vehicles, and Airbnb [18], which enables hosts to provide a visitor with a place to stay.

Despite the gradual rise in the quantity and variety of sharing-economy companies over the past several years [19], there not much research has been carried out on the evaluations and ratings that can be found on other sharing-economy platforms besides Airbnb. One exception is an investigation of BlaBlaCar, a car-sharing firm founded in France that has over one million registered users [20]. The research team chose Airbnb businesses to perform the field investigation. As a result of its success, the number of Airbnb research has lately increased [21], with the site accounting for approximately 75% of peer-to-peer lodging studies in the academic literature [22].

In 2008, Airbnb (Air bed and breakfast), as a type of sharing economy, started as a website where regular people can offer their vacant and unused rooms to travelers to rent as lodging facilities instead of traditional forms of tourism accommodation, such as hotels [23]. Currently, Airbnb has established its position as the biggest hospitality firm in the sharing economy, challenging the status quo [3].

Airbnb does not provide identical services to conventional hotels. It is known for offering unique and authentic customer experiences, making its guests feel "home away from home," which cannot be experienced in a traditional hotel [24]. Additionally, Airbnb provides cost leadership and also suggests a new way of life among local communities via interactive communication [25].

The notion that locals rent their houses informally to tourists is not a new trend. However, it increased due to new technology that facilitated virtual markets between hosts

and guests [26]. Another contributing factor was the emerging trend of fully independent tourists and the financial crisis [27]. Sharing lodging is a disruptive force in the economy, since it is less expensive, more practical, and simpler to use than standard hospitality services [27,28]. It has a good level of host participation [29] and might be considered similar to renting a house [24,30]. Moreover, lodging in the sharing economy could create an entirely new market [26]; Guttentag and Smith [31] revealed that, if the alternative had not been offered, 30 percent of Airbnb customers would not have traveled to the destinations or remained long. Various benefits can be achieved thanks to Airbnb. For example, local communities can benefit by enabling homeowners to earn income from their unused spaces, tourism can be expanded in areas less covered by traditional hotels, and new social ties can be created in addition to saving money for customers [32,33].

Bang Nong and Ha [34], Guttentag and Smith [31], and Heo et al. [35] demonstrated that Airbnb does not impact hospitality firms negatively, as it provides market opportunities for hotels and diversifies the supply by expanding the market to new categories of consumers [35]. Similarly, there is no rivalry between Airbnb and hotels because Airbnb targets a different market segments, such as vacation rentals, homestays, and young budget travelers [26]. Furthermore, several hospitality organizations have decided not to offer shared lodging because they do not believe that the sharing economy directly competes with them [35], but some are seriously worried and concerned [26]. For example, considering that sharing lodgings is always characterized by a limited cost, recreational visitors and hospitality organizations targeted toward a specific market could be badly affected [27,31,36].

Truong [37] suggests that Airbnb affects hotels positively. Since Airbnb can be perceived as a cheaper accommodation option, hotels may lower their prices to compete effectively, which, in turn, can bring more tourists to the destination. Furthermore, Airbnb increases business competitiveness, thus directing the hospitality industry to refine its products and enhance its appeal to consumers.

Given the benefits of Airbnb, various studies [13,38,39] were conducted to analyze the customers' decision-making process regarding using Airbnb. It was revealed that the key is sustainability and authentic experience from the local community [15]. However, the role of the perceived risks during the process is not yet fully considered [40].

On account of its intangible nature and heterogeneity, lodging is frequently viewed as a high-risk service [41]. For instance, the standard and quality of hospitality organizations may differ from one provider to the next, from one guest to another, and even from one day to the next [42]. Hence, customers have a greater degree of risk in their selection and have difficulty evaluating accommodations before purchasing [43].

*2.2. Risks*

Earlier studies on Airbnb mainly concentrated on customers to determine why they chose the service or how they were satisfied (e.g., [44,45]). The risk is conceptualized as a subjective feeling of uncertainty about the favorable outcome of the service or a subjective expectation of a potential loss [46]. Tourism and travel risks are associated with the probability of encountering difficulties while traveling [13,47]. Park et al. [48] described it as a customer's perception of possible uncertainty linked to harmful consequences in a buying process. As indicated by Cunningham [49] and Birkel [50], uncertainty and potential consequences have been described as the two components of risk. The former refers to an individual's subjective fear of losing something if their efforts do not produce favorable results, while the latter refers to their subjective feelings of certainty that the result would be undesirable [49,51].

A few researchers on tourism have made an effort to assess the effect of risk on intention and need. For example, Lee et al. [52] proposed an expanded approach that considers the fear of the influenza virus as a barrier that might prevent visitors from traveling to foreign nations. Although perceived risks were not accurately measured by the investigators, several researchers also considered prior experience as a factor in determining

intention and desire from the perspective that it reduces the perceived risk connected with choices concerning casino visits [53,54] or wine tour participation [55]. The rapid growth of Airbnb has brought optimism regarding its bright future in the industry and academia. However, consumers still experience various threats and risks [56].

Despite, Airbnb's rapid growth, its unique operating structure raised concerns [13,57]. The perceived risk of travelers using Airbnb is likely high because most travel experience relies on intangible, heterogeneous, and hard-to-standardize services [43,58–60]. High-involvement aspects of tourism services also increase the importance of perceived risk in the tourism industry [61]. Moreover, the perceived risk for travelers is likely high because most travel experiences and lodgings are frequently regarded as high-risk products because of their heterogeneity and intangibility [41]. Guests' decision-making processes are significantly influenced by perceived risk [62].

Additionally, Airbnb accommodation is a private space provided by individual hosts, it is difficult to maintain consistent quality levels following the manual [10]. Perceived risk works as an important factor in tourists' decision-making process [62].

Jacoby and Kaplan [63] proposed six major categories of losses: time, performance, financial, physiological, and physical losses. Jacoby and Kaplan's six constructs were also applied in this study to understand consumers' perceived risks toward Airbnb use. Performance risk is the mismatching between guests' expectations and the provided services [36,64–66]. It occurs when experiences at Airbnb places do not match the information posted on the Airbnb platform [67]. This risk exists because Airbnb has little control over the behavior of hosts and the absence of consistent hospitality standards applied to all Airbnb rentals [31].

The Financial risk relates to the potential for financial loss or not gaining the best possible monetary benefit [68]. Financial risk was considered the biggest threat to hospitality services [69,70], particularly for online purchased services [40].

By applying Jacoby and Kaplan's [63] definition, physical risk deals with safety and avoiding harm or injury during the stay. In comparison to hotels, Airbnb listings are unregulated and do not apply any strict safety requirements. It is up to hosts to ensure they provide a safe rental [71]. Consequently, one of the issues with using Airbnb is safety and personal security [67]. There are incidents involving physical risk. For example, one host gave his visitors drugs [72]. In another case, due to a damaged heater that filled the place with toxic gas, a Canadian woman died in an Airbnb rental in Taiwan [73].

The social risk depends on the notion that visiting an Airbnb accommodation could include social risk when friends or customers' relatives do not approve of their use of Airbnb [74]. Pires et al. [75] revealed that social risk was ranked as the second most significant risk by customers for both high- and low-engagement services bought online.

Psychological risk is the possible negative effect on a person's peace of mind caused by staying at an Airbnb [68,74]. Psychological risk dominates when consumers purchase an expensive, complex, or difficult-to-judge product [76]. Due to the intangibility of an Airbnb, in addition to paying in advance, it tends to have high psychological risk. Pires et al. [75] revealed that the psychological risk was rated as the highest for services bought online. Additionally, Airbnb consumers would make purchases through the company's mobile application and pay in advance. Based on user feedback on social media websites, refunding money can usually be complicated. [77].

Time risk includes wasting time or spending too much effort to book Airbnb accommodation [68,74]. Since Airbnb offers a large number of lodging options, it could be time-consuming for consumers to search for offers that are trustworthy [78].

Mitchell and Greatorex [79] and Hwang and Choe [80] revealed that time risk was the most significant risk for customers of hospitality. Additionally, it would take additional time and effort to contact the host before making a reservation because not all Airbnb properties use an instant booking option [26]. Potential visitors must log in, manage their profile, upload a photo of themselves, list their preferences, and send emails to reserve an Airbnb [81]. Following that, the Airbnb hosts choose to deny or accept the reservation [82].



## 3. Hypotheses Development

### 3.1. Risks and the Intention to Adopt Airbnb

The nature of Airbnb includes intangibility, heterogeneity, and inability to standardize its services, in addition to relying on an online platform to book it [41]. Furthermore, consumers' decision to use Airbnb instead of traditional hotels increased the level of uncertainty caused by less regulation and supervision of Airbnb compared to traditional hotels [10,83]. Thus, many risks may arise due to Airbnb conditions. Potential Airbnb customers are seriously concerned with risks when considering accommodation [31]. High-risk perception negatively affects travel intention. This was justified through the theory of reasoned action (TRA), which verified the relationship between perceived risk and the user's intentions [84,85]. According to Pavlou and Gefen [86], high degrees of perceived risk increase negative expectations, which create an unfavorable attitude that results in a negative influence on transaction intentions [87]. It can be proposed that:

**Hypothesis 1 (H1):** *Airbnb performance risk has a negative effect on tourists' intention to use Airbnb;*

**Hypothesis 2 (H2):** *Airbnb's financial risk has a negative effect on tourists' intention to use Airbnb;*

**Hypothesis 3 (H3):** *Airbnb's physical risk has a negative effect on tourists' intention to use Airbnb;*

**Hypothesis 4 (H4):** *Airbnb's social risk has a negative effect on tourists' intention to use Airbnb;*

**Hypothesis 5 (H5):** *Airbnb's time risk has a negative effect on tourists' intention to use Airbnb;*

**Hypothesis 6 (H6):** *Airbnb's psychological risk has a negative effect on tourists' intention to use Airbnb.*

### 3.2. Risks and the Intention to Adopt Airbnb in the Light of the Fear of COVID-19

The effect of risks on the decision-making process varies under different environmental changes concerning the level of uncertainty [88]. In times of stability, customers experience a good level of certainty and plan for change depending on their experience and previously acquired information [89,90]. However, the level of environmental changes caused by such a pandemic may make the decision-making process challenging due to the inability to predict changes [91].

To explain how people perceive various risks, a psychometric paradigm can be used. It consists of two components: cognitive and emotional risks. The first component depends on people's knowledge and controllability of the risk, whereas the latter relates to how risky the dangerous outcome (dread and immediacy) of the risk is [92]. According to Weber [93] and Zheng et al. [94], the fear function acts as a motivation factor to decrease or remove a feeling of risk and take action accordingly. In light of this approach, COVID-19 may increase or decrease the effect of the perceived risks on customers' decision-making.

Despite customers experiencing higher risks during the pandemic, their attitudes towards risks vary according to their ability to tolerate risks (risk aversion) [60,95]. For example, customers with high-risk aversion feel more threatened by ambiguous conditions, contrary to those with low-risk aversion [96]. Here, Chen [97] argued that tourists' risk aversion might affect their behavior after these risks occur. Thus, the fear of likely risk (COVID-19) and uncertainty instantly impact the tourist's intention to use Airbnb. It can be concluded that customers have different levels and types of sensitivity toward the risks of COVID-19. It can be proposed that:

**Hypothesis 7 (H7):** *The fear of COVID-19 negatively moderates the effect of Airbnb performance risk on tourists' intention to use Airbnb;*

**Hypothesis 8 (H8):** *The fear of COVID-19 negatively moderates the effect of Airbnb's financial risk on tourists' intention to use Airbnb;*

**Hypothesis 9 (H9):** *The fear of COVID-19 negatively moderates the effect of Airbnb's physical risk on tourists' intention to use Airbnb;*

**Hypothesis 10 (H10):** *The fear of COVID-19 negatively moderates the effect of Airbnb's social risk on tourists' intention to use Airbnb;*

**Hypothesis 11 (H11):** *The fear of COVID-19 negatively moderates the effect of Airbnb's time risk on tourists' intention to use Airbnb;*

**Hypothesis 12 (H12):** *The fear of COVID-19 negatively moderates the effect of Airbnb's psychological risk on tourists' intention to use Airbnb.*

## 4. Materials and Methods

### 4.1. Research Model Overview

The theoretical framework and relationships between the study variables are extracted from the extant literature. Figure 1 shows the study model that contains the suggested hypotheses, as follows:

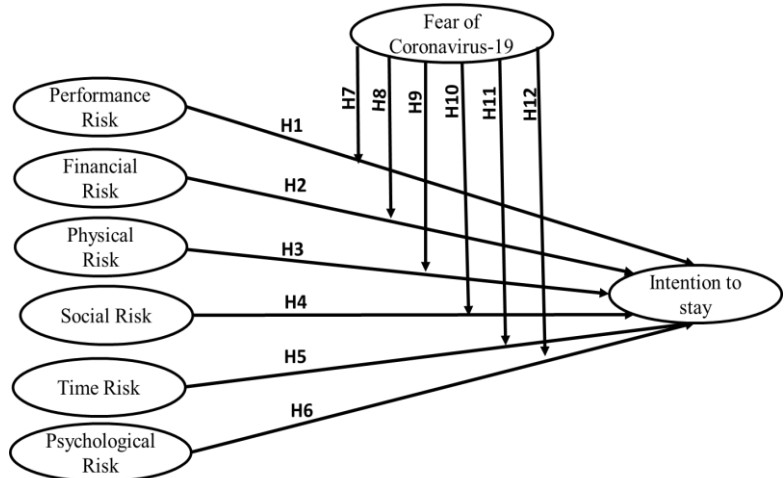

**Figure 1.** The proposed conceptual framework.

### 4.2. The Sample and Design

To investigate the proposed hypotheses, a questionnaire was developed, consisting of 31 questions regarding the studied variables "Airbnb risks, intention to stay, and fear of COVID-19", in addition to four questions to collect demographic data. Data was collected from customers of Airbnb properties in two tourist hotspot regions in Egypt, namely, Luxor and Aswan. These two distinct tourist areas were selected due to the substantial existence of Airbnb hosts in both cities.

This study employed a convenience sample for two reasons. First, to consider the geographical span of the study's Airbnb properties which are spread throughout two large tourist hotspots in Egypt. Second, because the sole requirement for this type of sampling approach is the participants' willingness to participate, and thus a simple random sample is not required.

While dealing with the surveys, the inspected properties were given confidentiality and anonymity while being surveyed. The researchers employed four persons, two in each city, to contact Airbnb customers who agreed to be surveyed. The data-collecting process took nearly five months (January to May 2021).

A total of 400 questionnaires were distributed to a convenience sample of the customers of Airbnb and 248 valid questionnaires were collected, with a recovery rate of 62%. As a general rule, Bagrowski and Gutowska [98] stated that "sample sizes of 200 to

300 respondents provide an acceptable margin of error and fall before the point of diminishing returns". Therefore, the sample size of 248 was considered appropriate.

Regarding the features of the sample, this sample consisted of 69.4% males and 30.6% females, their ages ranging from 24 to 61 years old. Approximately two-thirds of respondents (n = 162, 65.3%) completed postgraduate studies. The majority of respondents (n = 242, 97.6%) were married.

### 4.3. Measures

The 21-item scale of Jun [74] was used to measure Airbnb risks; four items for performance risk, four items for financial risk, three items for physical risk, three items for social risk, four items for time risk, and three items for psychological risk. A 3-item scale derived from Chen and Chang [99] was used to assess the intention to use Airbnb. The fear of COVID-19 was rated by Airbnb customers using a seven-item scale according to Ahorsu et al. [100]. All items were measured on a five-point Likert scale ranging from disagree (1) to agree (5). The questionnaire survey is attached in Appendix A.

## 5. Results

This current research utilized the Structure Equation Model (SEM) via the Partial Least Squares (PLS) technique to test the hypotheses of the study with SmartPLS-4.0. The proposed theoretical model was examined using a two-step approach suggested by Leguina [101], As follows.

### 5.1. Assessment of outer Measurement Model

To evaluate the outer model's reliability and validity, internal consistency reliability, indicator reliability, convergent validity, and discriminant validity were tested. First, as displayed in Table 1, the structures' internal consistency reliability was tested with Cronbach's alpha ($\alpha$), changing from 0.774 to 0.928, and the composite reliability (CR) ranging from 0.869 to 0.953. Second, the indicators' reliability was acceptable, as all loading values of the structure indicators were higher than 0.70. Third, convergent validity was evaluated by the average variance extracted (AVE) values exceeding the satisfactory value of 0.50 [102].

**Table 1.** Assessment of the formative measurement model.

| The Model Items | Outer Loading | $\alpha$ | C.R | AVE |
|---|---|---|---|---|
| **Performance Risk** | | 0.868 | 0.909 | 0.714 |
| Perf_Risk_1 | 0.831 | | | |
| Perf_Risk_2 | 0.870 | | | |
| Perf_Risk_3 | 0.866 | | | |
| Perf_Risk_4 | 0.811 | | | |
| **Financial Risk** | | 0.823 | 0.883 | 0.654 |
| Fina_Risk_1 | 0.741 | | | |
| Fina_Risk_2 | 0.736 | | | |
| Fina_Risk_3 | 0.876 | | | |
| Fina_Risk_4 | 0.871 | | | |
| **Physical Risk** | | 0.860 | 0.912 | 0.776 |
| Physical_Risk_1 | 0.814 | | | |
| Physical_Risk_2 | 0.928 | | | |
| Physical_Risk_3 | 0.897 | | | |
| **Social Risk** | | 0.774 | 0.869 | 0.689 |
| Social_Risk_1 | 0.754 | | | |
| Social_Risk_2 | 0.899 | | | |
| Social_Risk_3 | 0.831 | | | |

**Table 1.** *Cont.*

| The Model Items | Outer Loading | α | C.R | AVE |
|---|---|---|---|---|
| **Time Risk** | | 0.917 | 0.932 | 0.775 |
| Time_Risk_1 | 0.933 | | | |
| Time_Risk_2 | 0.950 | | | |
| Time_Risk_3 | 0.846 | | | |
| Time_Risk_4 | 0.782 | | | |
| **Psychological Risk** | | 0.928 | 0.953 | 0.872 |
| Psyc_Risk_1 | 0.931 | | | |
| Psyc_Risk_2 | 0.951 | | | |
| Psyc_Risk_3 | 0.919 | | | |
| **Intention to stay** | | 0.901 | 0.938 | 0.834 |
| Inte_Stay_1 | 0.938 | | | |
| Inte_Stay_2 | 0.941 | | | |
| Inte_Stay_3 | 0.858 | | | |
| **Fear of COVID-19.** | | 0.901 | 0.920 | 0.622 |
| Fear_Corona_1 | 0.704 | | | |
| Fear_Corona_2 | 0.855 | | | |
| Fear_Corona_3 | 0.791 | | | |
| Fear_Corona_4 | 0.767 | | | |
| Fear_Corona_5 | 0.720 | | | |
| Fear_Corona_6 | 0.848 | | | |
| Fear_Corona_7 | 0.820 | | | |

Finally, three criteria were implemented to assess the discriminant validity of the constructs. They were cross-loading, Fornell–Larcker criterion, and heterotrait–monotrait ratio (HTMT) [101,103]. As indicated in Table 2, the outer loading for each latent variable bolded was higher than the cross-loading with other measurements.

**Table 2.** Cross loading results.

| | 1 | 2 | 3 | 4 | 5 | 6 | 7 | 8 |
|---|---|---|---|---|---|---|---|---|
| Fear_Corona_1 | **0.704** | −0.128 | −0.135 | 0.093 | −0.118 | 0.002 | 0.271 | −0.018 |
| Fear_Corona_2 | **0.855** | 0.065 | −0.295 | 0.217 | 0.083 | 0.111 | 0.399 | 0.017 |
| Fear_Corona_3 | **0.791** | −0.031 | −0.169 | −0.004 | 0.143 | −0.011 | 0.330 | 0.185 |
| Fear_Corona_4 | **0.767** | 0.012 | −0.170 | −0.015 | 0.108 | 0.079 | 0.410 | 0.070 |
| Fear_Corona_5 | **0.720** | 0.040 | −0.147 | −0.055 | 0.053 | 0.014 | 0.129 | 0.104 |
| Fear_Corona_6 | **0.848** | 0.160 | −0.295 | 0.024 | 0.172 | 0.182 | 0.408 | 0.095 |
| Fear_Corona_7 | **0.820** | 0.042 | −0.135 | −0.120 | 0.190 | −0.072 | 0.255 | −0.086 |
| Fina_Risk_1 | −0.071 | **0.741** | −0.192 | 0.203 | 0.246 | −0.075 | 0.051 | 0.108 |
| Fina_Risk_2 | −0.040 | **0.736** | −0.210 | 0.069 | 0.357 | −0.098 | 0.098 | 0.098 |
| Fina_Risk_3 | 0.168 | **0.876** | −0.267 | 0.164 | 0.263 | −0.055 | 0.055 | 0.006 |
| Fina_Risk_4 | 0.068 | **0.871** | −0.260 | 0.133 | 0.292 | −0.062 | 0.080 | −0.004 |
| Inte_Stay_1 | −0.282 | −0.269 | **0.938** | −0.386 | −0.316 | −0.216 | −0.485 | −0.212 |
| Inte_Stay_2 | −0.320 | −0.279 | **0.941** | −0.378 | −0.334 | −0.225 | −0.485 | −0.324 |
| Inte_Stay_3 | −0.084 | −0.245 | **0.858** | −0.285 | −0.148 | −0.168 | −0.409 | −0.146 |
| Perf_Risk_1 | 0.075 | 0.238 | −0.227 | **0.831** | 0.085 | −0.002 | 0.247 | −0.080 |
| Perf_Risk_2 | 0.088 | 0.267 | −0.425 | **0.870** | 0.070 | −0.019 | 0.322 | 0.109 |
| Perf_Risk_3 | −0.065 | 0.033 | −0.293 | **0.866** | 0.004 | −0.032 | 0.176 | 0.015 |
| Perf_Risk_4 | 0.064 | 0.025 | −0.307 | **0.811** | −0.059 | 0.095 | 0.282 | −0.101 |
| Physical_Risk_1 | −0.144 | 0.260 | −0.186 | −0.038 | **0.814** | −0.183 | 0.086 | 0.041 |
| Physical_Risk_2 | 0.095 | 0.379 | −0.262 | −0.034 | **0.928** | −0.201 | 0.099 | 0.051 |
| Physical_Risk_3 | 0.271 | 0.294 | −0.325 | 0.114 | **0.897** | −0.156 | 0.296 | −0.083 |
| Psyc_Risk_1 | 0.043 | −0.092 | −0.211 | 0.065 | −0.146 | **0.931** | 0.357 | 0.088 |
| Psyc_Risk_2 | 0.067 | −0.036 | −0.243 | −0.019 | −0.187 | **0.951** | 0.271 | 0.189 |
| Psyc_Risk_3 | 0.142 | −0.135 | −0.163 | −0.019 | −0.241 | **0.919** | 0.306 | 0.041 |

**Table 2.** *Cont.*

|  | 1 | 2 | 3 | 4 | 5 | 6 | 7 | 8 |
|---|---|---|---|---|---|---|---|---|
| Social_Risk_1 | 0.443 | 0.170 | −0.334 | 0.284 | 0.388 | 0.210 | **0.754** | 0.134 |
| Social_Risk_2 | 0.352 | 0.081 | −0.462 | 0.319 | 0.180 | 0.329 | **0.899** | 0.241 |
| Social_Risk_3 | 0.283 | −0.009 | −0.450 | 0.184 | −0.007 | 0.272 | **0.831** | 0.237 |
| Time_Risk_1 | 0.075 | 0.058 | −0.271 | −0.071 | −0.057 | 0.112 | 0.259 | **0.933** |
| Time_Risk_2 | 0.075 | 0.054 | −0.289 | 0.037 | 0.029 | 0.178 | 0.258 | **0.950** |
| Time_Risk_3 | 0.041 | 0.025 | −0.070 | 0.047 | −0.059 | 0.090 | 0.125 | **0.846** |
| Time_Risk_4 | 0.025 | 0.037 | −0.087 | 0.073 | 0.061 | −0.085 | 0.133 | **0.782** |

1 → Fear of COVID-19; 2 → Financial Risk; 3 → Intention to stay; 4 → Performance Risk; 5 → Physical Risk; 6 → Psychological Risk; 7 → Social Risk; 8 → Time Risk.

As shown in Table 3, the bolded values of the AVEs in the diagonals are higher than the correlation between variables. According to Gold et al. [104], HTMT values need to be less than 0.90. The study's values of HTMT were lower than this (Table 4). According to the results, the model structure has adequate discriminant validity. Consequently, the outer measurement model outcomes were deemed strong enough to continue to evaluate the structural model.

**Table 3.** The square root of AVE.

|  | 1 | 2 | 3 | 4 | 5 | 6 | 7 | 8 |
|---|---|---|---|---|---|---|---|---|
| **Fear of COVID−19** | **0.788** | | | | | | | |
| **Financial Risk** | 0.053 | **0.809** | | | | | | |
| **Intention to stay** | −0.268 | −0.290 | **0.913** | | | | | |
| **Performance Risk** | 0.052 | 0.174 | −0.389 | **0.845** | | | | |
| **Physical Risk** | 0.126 | 0.355 | −0.305 | 0.030 | **0.881** | | | |
| **Psychological Risk** | 0.084 | −0.087 | −0.225 | 0.011 | −0.200 | **0.934** | | |
| **Social Risk** | 0.421 | 0.087 | −0.507 | 0.312 | 0.201 | 0.331 | **0.830** | |
| **Time Risk** | 0.072 | 0.055 | −0.259 | 0.002 | −0.009 | 0.123 | 0.252 | **0.880** |

**Table 4.** HTMT results.

|  | 1 | 2 | 3 | 4 | 5 | 6 | 7 | 8 |
|---|---|---|---|---|---|---|---|---|
| **Fear of COVID−19** | 0.788 | | | | | | | |
| **Financial Risk** | 0.053 | 0.809 | | | | | | |
| **Intention to stay** | −0.268 | −0.290 | 0.913 | | | | | |
| **Performance Risk** | 0.052 | 0.174 | −0.389 | 0.845 | | | | |
| **Physical Risk** | 0.126 | 0.355 | −0.305 | 0.030 | 0.881 | | | |
| **Psychological Risk** | 0.084 | −0.087 | −0.225 | 0.011 | −0.200 | 0.934 | | |
| **Social Risk** | 0.421 | 0.087 | −0.507 | 0.312 | 0.201 | 0.331 | 0.830 | |
| **Time Risk** | 0.072 | 0.055 | −0.259 | 0.002 | −0.009 | 0.123 | 0.252 | 0.880 |

*5.2. Assessment of the Structural Model*

The hypotheses were then tested by a structural equation analysis. In particular, the model's predictive capacity and explanatory power were analyzed [105]. With the VIF values of the manifest indicators changing from 1.000 to 3.977 below 5, the multicollinearity of the structural model has been verified as inexistent [106]. Next, Chin, [107] indicated that the lower limit for the $R^2$ values is 0.10. Therefore, the $R^2$ values for the variables of intention to stay being 0.585are acceptable (Table 5). Additionally, The Stone–Geisser $Q^2$ test indicates that the intention to stay value is greater than zero (Table 5), providing adequate

predictive validity for the model [102,108]. Accordingly, enough predictive validity for the structural model was also confirmed.

**Table 5.** Coefficient of determination ($R^2$) and ($Q^2$) of the model.

| Endogenous Latent Construct | ($R^2$) | ($Q^2$) |
|:---:|:---:|:---:|
| **Intention to stay** | 0.585 | 0.440 |

Lastly, the path coefficient and t-value of the hypothesized association were analyzed using a bootstrapping technique. Table 6 and Figure 2 display the hypothesis test results, given the path coefficient values and the relevant significance. Financial, performance, physical, psychological, social, and time risks variables were negative, while there was a significant value indicating the intention to stay in an Airbnb, respectively (β = −0.172, $p < 0.000$; β =−0.137, $p < 0.014$; β = −0.202, $p < 0.000$; β = −0.220, $p < 0.001$; β =−0.222, $p < 0.001$; β = −0.139, $p < 0.027$), meaning that H1, H2, H3, H4, H5, and H6 were supported. The results also showed that the moderating effect of fear of COVID-19 on the financial risks variable concerning the intention to stay was insignificant (β =0.006, $p < 0.922$), disproving H7. However, results confirmed the moderating effect of fear of COVID-19 variable on physical, psychological, social, and time risks variables on the intention to stay at an Airbnb (β = 0.345, $p < 0.000$; β = −0.186, $p < 0.003$; β = −0.343, $p < 0.000$; β = −0.291, $p < 0.000$; β = −0.223, $p < 0.041$), supporting H8, H9, H10, H11, H12.

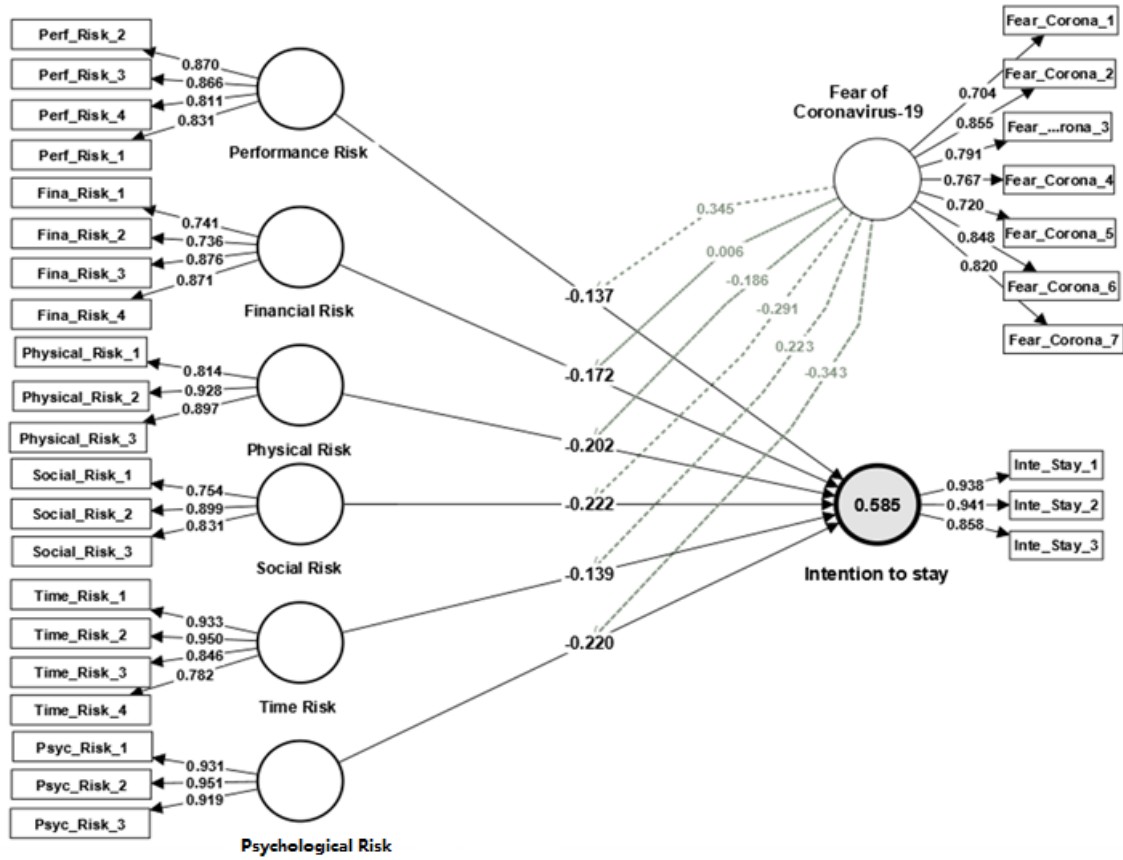

**Figure 2.** The study's Inner and Outer model.

**Table 6.** Path Coefficients.

|  |  | Beta (β) | t-Values | *p*-Values | Results |
|---|---|---|---|---|---|
| **H1** | Financial Risk → Intention to stay | −0.172 | 4.181 | 0.000 | Accepted |
| **H2** | Performance Risk → Intention to stay | −0.137 | 2.475 | 0.014 | Accepted |
| **H3** | Physical Risk → Intention to stay | −0.202 | 3.545 | 0.000 | Accepted |
| **H4** | Psychological Risk → Intention to stay | −0.220 | 3.445 | 0.001 | Accepted |
| **H5** | Social Risk → Intention to stay | −0.222 | 3.424 | 0.001 | Accepted |
| **H6** | Time Risk → Intention to stay | −0.139 | 2.222 | 0.027 | Accepted |
| **H7** | Moderating Effect 1(Financial Risk × Fear of COVID-19) → Intention to stay | 0.006 | 0.098 | 0.922 | Not Accepted |
| **H8** | Moderating Effect 2 (Performance Risk × Fear of COVID-19) → Intention to stay | 0.345 | 5.575 | 0.000 | Not Accepted |
| **H9** | Moderating Effect 3 (Physical Risk × Fear of COVID-19) → Intention to stay | −0.186 | 2.983 | 0.003 | Accepted |
| **H10** | Moderating Effect 4 (Psychological Risk × Fear of COVID-19) → Intention to stay | −0.343 | 3.648 | 0.000 | Accepted |
| **H11** | Moderating Effect 5 (Social Risk × Fear of COVID-19) → Intention to stay | −0.291 | 3.735 | 0.000 | Accepted |
| **H12** | Moderating Effect 6 (Time Risk × Fear of COVID-19) → Intention to stay | 0.223 | 2.472 | 0.014 | Not Accepted |

## 6. Discussion

### 6.1. Airbnb Risks and Intention to Stay in Airbnb

The empirical results of this study revealed that financial, performance, physical, psychological, social, and time risks have a direct and significant negative influence on the intention to stay at an Airbnb. This agrees with previous studies regarding the domain of customer purchasing decisions (e.g., [60,109]). These risks may arise due to several factors encompassing the intangible nature of services provided in the context of hospitality and tourism including Airbnb, where neither guests nor service providers can examine physically what they sell and buy. Consequently, the level of uncertainty dominates the process [110]. Additionally, Airbnb risks may be due to the mechanism of booking that takes place on the website of Airbnb or through an online application that anyone can access, offering a different experience that may not be available in other accommodation-booking options. Consumers may vary in interpreting this experience, due to the subjectivity of travel and tourism experiences [111]. Similarly, Airbnb is described as having uncertain quality and an uncertain image because it depends mainly on individuals with little professional training and experience to provide accommodation services [112]. The aforementioned reasons may increase the level of uncertainty and thus risks experienced in dealing with Airbnb, which negatively affect tourists' intention to use Airbnb as an accommodation means [113].

### 6.2. Assessing the Moderating Effect

Regarding the indirect relationship between Airbnb's different risks and intention to stay at an Airbnb moderated by the level of fear of COVID-19, it was indicated that the moderating effect of fear of COVID-19 between financial risk and the intention to stay is insignificant. This is in line with the study of So et al.'s [44], who revealed that the perceived risk had no significant relationship to attitude or behavioral intentions.

On the one hand, the moderating effect of fear of COVID-19 between Airbnb performance and time risk and the intention to stay at an Airbnb is significant but positive. According to Figures 3 and 4, we conclude that the fear of COVID-19 variable can dampen the negative relationship between Airbnb's performance, time risks, and the intention to stay at an Airbnb. This means that, when tourists experience a high level of fear of COVID-19, they tend to tolerate Airbnb performance and time risks and tend to stay at an Airbnb.

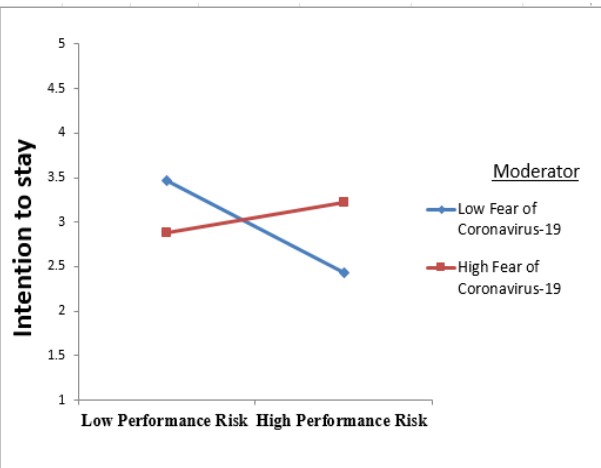

**Figure 3.** Interaction plot for the moderating effect of fear of COVID-19 between performance risk and intention to stay at an Airbnb.

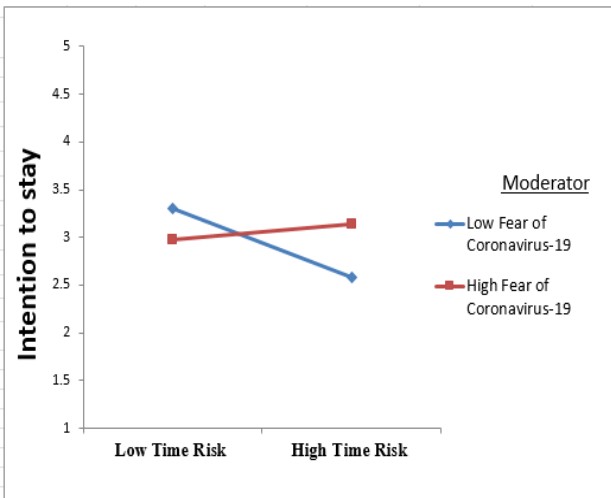

**Figure 4.** Interaction plot for the moderating effect of fear of COVID-19 between time risk and intention to stay at an Airbnb.

On the other hand, the fear of COVID-19 can strengthen the negative relationship between Airbnb's physical, psychological, and social risks and the intention to stay at an Airbnb (see Figures 5–7). The results also reveal the moderating effect of the fear of COVID-19, the results corroborating the findings by Mohsin and Lengler [114].

To explain these results, prospect theory, which is widely used to illustrate human behavior, can be adopted. According to prospect theory, developed by Kahneman and Tversky [115] to explain the decision-making process in the conditions of risk and uncertainty, it is suggested that a person's behavior depends upon two aspects: the result of the action (i.e., gains or loss) and risk attitudes. The perceived risks stand for the expected loss, while the perceived value reflects the benefits after comparing the gains and sacrifices that come with pursuing a desired outcome [116]. Generally, people are averse to losses, so the perceived value has a positive effect on customer behavior. In parallel with prospect theory, Kock et al. [117] suggested that tourists' behavioral changes occurred during and after the COVID-19 period that helped tourists adapt to the conditions of COVID-19 and achieve self-protection and disease avoidance. Consequently, in light of the fear of COVID-19, it was safer to rent a property offered by Airbnb compared to booking a hotel room, as Airbnb decreases direct person-to-person contact, thus reducing the spread of the coronavirus [118]. Thus, having a high level of fear of COVID-19 encourages tourists to stay at

Airbnbs, despite its time and performance risks. However, they will not tolerate other risks where enhancements must be carried out.

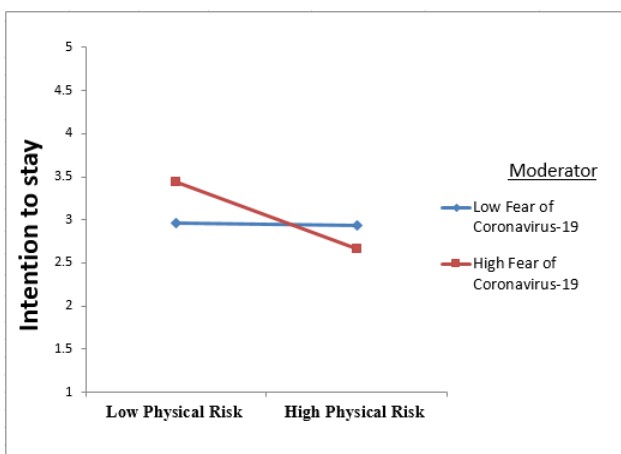

**Figure 5.** Interaction plot for the moderating effect of fear of COVID-19 between physical risks and intention to stay at an Airbnb.

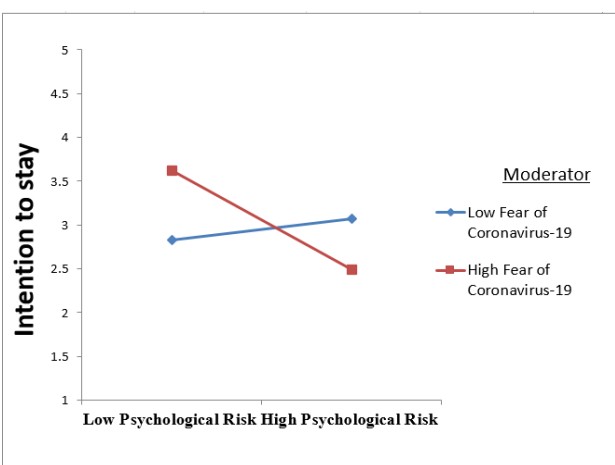

**Figure 6.** Interaction plot for the moderating effect of fear of COVID-19 between psychological risk and intention to stay at an Airbnb.

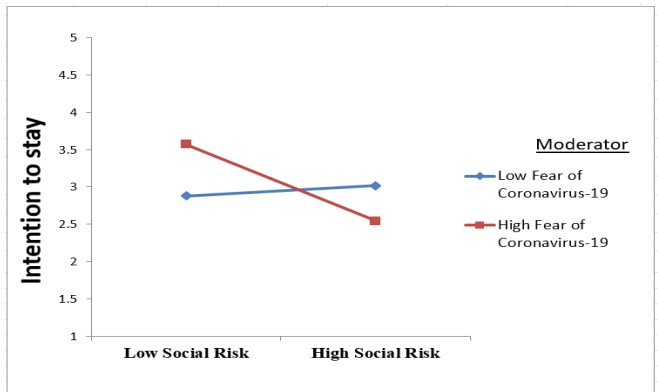

**Figure 7.** Interaction plot for the moderating effect of fear of COVID-19 between social risk and intention to stay at an Airbnb.

## 7. Conclusions and Recommendations

This study shed light on the direct effect of risks on tourists' intention to use Airbnb and the indirect effect in light of the fear of COVID-19. Generally, risks have direct negative effects on tourists' intention to use Airbnb. However, the fear of COVID-19 reshaped this effect (i.e., tourists tolerate performance and time risks and prefer to use Airbnb when the fear of COVID-19 is high. Conversely, the fear of COVID-19 did not modify the negative effect of other types of risks on Airbnb use).

It is suggested that Airbnb has an edge over hotels under the conditions of fear of COVID-19, as it is considered a safe place that provides less interaction among tourists, thus decreasing the opportunities for infection. However, the essence of other types of risks needs to be overcome.

Airbnb needs to address the types of risks experienced by its guest to expand widely. Airbnb needs to improve its guest trust by providing more information about the host, including operating experience, response rate, and host status. To decrease the level of risk related to using Airbnb, it is proposed to provide more information regarding the product and the provider of an Airbnb and prompt follow-up if any issues arise. This can enhance trust, consequently decreasing the perceived risk.

It's recommended that Airbnb hosts keep up with the professionalism involved in providing unique experiences for their customers. Some possible actions, such as effective security features and emergency lines and services, can minimize the perceived risks of using Airbnb.

The Ministry of Tourism and Antiquities should launch initiatives to increase awareness of Airbnb hosts and the effect of Airbnb on Egypt as a tourist destination. Additionally, Airbnb activities should be supervised formally. In a similar vein, Airbnb activities should not be conducted on an individual basis. Instead, a legislative framework should control it.

## 8. Limitations and Future Research

The findings of the study should be considered in light of its limitations. Firstly, this study used a convenience sample, indicating that the results should be adapted with caution. Consequently, using a larger sample size is proposed. Future studies should also involve longitudinal studies because they can provide various results depending on the sequence of events over time. Secondly, future studies may address the various risks perceived in using Airbnb and how to overcome them. Finally, future studies could conduct multi-group analyses to identify how the results vary according to different types of accommodations (e.g., private rooms, shared rooms, and entire homes).

**Author Contributions:** Conceptualization, S.F., O.E.S. and M.F.A.; data curation, S.F., O.E.S., M.F.A. and N.A.; formal analysis, S.F., M.F.A. and H.A.K.; funding acquisition, N.A.; investigation, S.F., M.F.A. and H.A.K.; methodology, S.F., O.E.S., M.F.A. and H.A.K.; project administration, O.E.S., S.F. and M.F.A.; resources, S.F., N.A., M.F.A. and O.E.S.; software H.A.K. and N.A.; supervision, S.F., O.E.S. and M.F.A.; validation, S.F., O.E.S., N.A. and M.F.A.; visualization, S.F., N.A., M.F.A. and H.A.K. writing—original draft, M.F.A., S.F. and O.E.S.; writing—review and editing, S.F., O.E.S., M.F.A., H.A.K. and N.A. All authors have read and agreed to the published version of the manuscript.

**Funding:** This research received no external funding.

**Institutional Review Board Statement:** Not applicable.

**Informed Consent Statement:** Informed consent was obtained from all subjects involved in the study.

**Data Availability Statement:** Data is available upon request from researchers who meet the eligibility criteria. Kindly contact the first author privately through e-mail.

**Conflicts of Interest:** The authors declare no conflict of interest.

## Appendix A

Airbnb risks

Performance Risk

- I worry that the Airbnb place/listing will not match the photos posted online.
- I worry that the Airbnb place/listing will not match the descriptions posted online.
- I worry that the Airbnb place/listing would not be clean.
- I am concerned that the Airbnb host would treat me unkindly.

Financial Risk

- I am concerned about whether an Airbnb place is more expensive than hotel rooms in the same area.
- I am concerned that an Airbnb place is overpriced considering the quality.
- I am concerned if the price of an accommodation on the Airbnb website is more expensive compared to other travel websites.
- I am concerned that staying at an Airbnb place could involve financial losses.

Physical Risk

- I am concerned that staying at an Airbnb place/listing would lead to something bad happening to me.

- I worry that the Airbnb host may do something bad to me.
- I am concerned that it may not be safe to stay at an Airbnb place/listing.

Social Risk

- Staying at an Airbnb place/listing will adversely affect others' opinion of me.
- I would be thought of as foolish by people whose opinion I value if I stay at an Airbnb place/listing.
- The thought of staying at an Airbnb place/listing causes me concern because some friends would think I was just being showy.

Time Risk

- Using Airbnb to book accommodation takes a lot of effort.
- Using Airbnb to book accommodation will be a waste of time.
- Using Airbnb to book accommodation will take too much time.
- Using Airbnb could lead to an inefficient use of my time as I have to send messages to the host.

Psychological Risk

- The thought of staying at an Airbnb place/listing makes me feel psychologically uncomfortable.
- The thought of staying at an Airbnb place/listing gives me a feeling of unwanted anxiety.
- The thought of staying at an Airbnb place/listing causes me to experience unnecessary tension.

**Fear of Coronavirus-19 Scale**

- I am most afraid of coronavirus-19.
- It makes me uncomfortable to think about coronavirus-19.
- My hands become clammy when I think about coronavirus-19.
- I am afraid of losing my life because of coronavirus-19.
- When watching news and stories about coronavirus-19 on social media, I become nervous or anxious.
- I cannot sleep because I'm worrying about getting coronavirus-19.
- My heart races or palpitates when I think about getting coronavirus-19.

**Intention to revisit**

- I will stay at Airbnb places.
- I will recommend my friends to stay at Airbnb places.
- When I make decision on accommodation, I will choose Airbnb places.

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
