# Peer review of "Risks in Relation to Adopting Airbnb Accommodation: The Role of Fear of COVID-19"

_sustainability, doi:10.3390/su15065050_

Round 1
Reviewer 1 Report
-The previous works on the topic is not investigated appropriately.
-it is not clear how the data collection is performed. A good description on the procedure is needed.
-The theory for the development of the model must be explained.
-Introduction section is poorly presented. The related works are ignored by the authors.
-The majority of the hypotheses are developed by old references.
-The questionnaire survey must be attached in the appendix.
Reviewer 2 Report
Dear Author
Thank you for the opportunity of reading your work. Although I have a few questions to share with you, in order to improve your articles quality.
Abstract
Please operaceionalize your objetives
Introduction
You stated on line 40 that Covid-19 affects tourism and.....
How? Your should show facts with sources to this affirmation.
Last paragraph of Introduction should present the articles structure.
1. Literature review:
Should start with Renting Short Term Accommodation, state of art
Should presents other examples, beside Airbnb
2.2 Airbnb and hotels:
Why you have and hotels? You don't talk about them, even to show the main differences to Airbnb
You said on line 77: Some academics? Who are they?
Should not talk about opinions, you are in Literature review
2.3. Risks
You should present the concept of risk and risk adopted to tourism
line 98 missed References
Why you choose Jacoby and Kaplan constructs?
3. Hypotheses development
references are missing
Risk aversion? where this information is descripted on Literature Review? Please add.
4.2. Sample and design? How 33 questions are divided? How the questionnaire was built? Questions are about what? open or closed questions?
5. Results
SEM? Why don't´you made a reference or explanation in the methods part?
6. Discussion?
Where is the comparison of your results and other authors results?
7. Conclusion?
Where are the contributions for the academy and for professionals?
8. Future research
FR should be better developed
Reviewer 3 Report
Thank you for allowing me to review this manuscript. I have several suggestions to improve the manuscript:
Abstract
Please, rewrite the first three sentences in the abstract.
P.1 line 15. 'This study aims ……….'
P.1 line 16 'The research had two aims:……..'?????
P.1 line 17-18 has grammatical errors.
P.1 line 24-25 delete the following sentences 'Contradictory, the fear of COVID-19 didn't moderate the relationship between Airbnb (performance and time risks) and the intention to adopt Airbnb,' and combine the rest with the previous sentence.
Add a sentence that covers limitations and future suggestions.
Introduction
Starting from the first sentences and continuing until the end of the manuscript, several sentences need in-text references or citations (e.g., page 1, lines 36–38, etc.).
P.2 line 53, the authors should extend the last paragraph or add a paragraph related to:
How can these tested relations contribute to the tourism field? Briefly, I suggest the authors write a captivating introduction, including the research problem, what the gap is, how this study fills the gap and the contribution of the study or why you are doing this research.
Literature Review
P.2 line 88, give or cite some references or studies (in-text cite) for 'various studies' what are those studies? Let us know.
Don't understand why we need the section 'Airbnb and hotels' section.
P.2, line 94 needs a reference and page number because you are making a definition.
Please add also cite for lines 98-100; 107;113; 121; 125. Also, use capital letters for lines 115 and 120 for 'physical' and 'psychological'.
The literature is better than the introduction; however, it is short. The author(s) should write more about how other scholars tested those relationships and what they found. They should share previous scholars' studies or findings as an example instead of just citing them.
Methods:
When did you collect the data and give more demographic info about the sample?
Why did you prefer to use convenience sampling? Please clarify.
Give more information about the study site.
My major concern is that 248 respondents are not adequate for the whole of Egypt.
I wish the author had chosen a specific destination or city in Eygpt.
Results
Please create a fit indices table covering IFI, TLI, and CFI scores for CFA and SEM.
Conclusion
In other words, the last section is not elucidated enough. There should be a discussion of rationality for the outcome, and the author(s) should discuss the results of the studies and compare them to previous studies that have similar or opposing results. As a final suggestion, the limitations and implications of the study are too short. Especially, I would suggest the author(s) improve the practical implication section. Overall, the study's contribution to theory and practice is not clear, noticeable, or evident. I wish the author the best of luck with the revision.
Round 2
Reviewer 1 Report
There are many writing errors in this work which must be fixed. Example:
to figure 5, 6, and 7. the results also validate moderation influences 403 of fear of COVID-19, the results corroborate the findings by Mohsin and Lengler
Reviewer 2 Report
Dear author
Thank you for the improvements. Please pay attention to Hypothesis number 6. Numer 5 is repeated.
Thank you
Reviewer 3 Report
Now, it is much better and I believe the manuscript has been sufficiently improved to publish in Sustainability.
